# Metabolic Singularities in Microsatellite-Stable Colorectal Cancer: Identifying Key Players in Immunosuppression to Improve the Immunotherapy Response

**DOI:** 10.3390/cancers17030498

**Published:** 2025-02-02

**Authors:** Teresa Gorría, Marina Sierra-Boada, Mariam Rojas, Carolina Figueras, Silvia Marin, Sergio Madurga, Marta Cascante, Joan Maurel

**Affiliations:** 1Medical Oncology Department, Hospital Clínic de Barcelona, 08036 Barcelona, Spain; tgorria@clinic.cat (T.G.); rojasp@recerca.clinic.cat (M.R.); cfiguera@clinic.cat (C.F.); 2Translational Genomics and Targeted Therapies in Solid Tumors, Agustí Pi i Sunyer Biomedical Research Institute (IDIBAPS), 08036 Barcelona, Spain; 3Medicine Department, University of Barcelona, 08036 Barcelona, Spain; 4Medical Oncology Department, Parc Taulí Hospital Universitari, Institut d’Investigació i Innovació Parc Taulí (I3PT-CERCA), Universitat Autònoma de Barcelona, 08208 Sabadell, Spain; msierra@tauli.cat; 5Department of Biochemistry and Molecular Biomedicine, University of Barcelona, 08036 Barcelona, Spain; silviamarin@ub.edu; 6Institute of Biomedicine of University of Barcelona (IBUB), 08036 Barcelona, Spain; 7Centro de Investigación Biomédica en Red de Enfermedades Hepáticas y Digestivas (CIBEREHD), Instituto de Salud Carlos III (ISCIII), 28029 Madrid, Spain; 8Department of Material Science and Physical Chemistry, Research Institute of Theoretical and Computational Chemistry (IQTCUB), University of Barcelona, 08028 Barcelona, Spain; s.madurga@ub.edu

**Keywords:** colorectal cancer, immunotherapy, metabolism, tumor microenvironment, microsatellite stability

## Abstract

Immunotherapy is the standard treatment for microsatellite-unstable metastatic colorectal cancer. However, it remains ineffective in microsatellite-stable colorectal cancer, which is representative most of cases. This review explores the metabolic singularities, including transcriptomic, proteomic, and post-translational modifications, that drive immunosuppression in microsatellite-stable colorectal cancer. We discuss the limited efficacy of immunotherapy in microsatellite-stable colorectal cancer and emphasize the importance of identifying robust biomarkers to predict therapeutic response. Additionally, we describe recent findings that uncover unique metabolic profiles in colorectal cancer tumors and immune cells, emphasizing the significant impact of metabolic interactions on immune evasion mechanisms.

## 1. Introduction

Colorectal cancer (CRC) is the third most frequently diagnosed cancer globally and the second leading cause of cancer-related death [1,2]. Driven by population aging and growth, projections suggest that by 2035 deaths from colon cancer and rectal cancer will have surged by 60.0% and 71.5%, respectively [3]. Advances in the early diagnostic methods and therapeutic strategies for both localized and advanced CRC remain a critical and ongoing challenge, particularly because patients with metastatic CRC (mCRC) face a poor prognosis, with a 5-year survival rate of less than 15% [4].

In CRC, surgery, radiotherapy (RT), and chemotherapy play fundamental roles in the therapeutic approach. However, in recent years, the introduction of immunotherapy with immune checkpoint inhibitors (ICIs) has revolutionized the treatment of various types of tumors, such as melanoma and lung cancer, significantly improving both survival rates and treatment responses and becoming the cornerstone of treatment [5,6,7]. The impact of immunotherapy in CRC has not met expectations, except in the subset of patients with mismatch repair-deficient (MMRd) or microsatellite-instability-high (MSI-H) CRC, in whom a favorable response to ICIs has been observed [8,9]. This is likely due to the association of these tumors with greater immune infiltration and an increased number of neoantigens. However, this subgroup represents only about 15% of CRC patients and less than 5% of those with mCRC, indicating that most CRC cases are mismatch repair proficient (MMRp)/microsatellite stable (MSS). This predominant MMRp/MSS population has shown little to no response to ICIs. Consequently, the following are emerging and highly relevant areas of research: understanding the factors contributing to immunotherapy resistance, exploring new mechanisms to overcome this resistance in CRC, and identifying novel predictive biomarkers of efficacy. In this review, we discuss the impact of immunotherapy in CRC, highlighting the differences between the MSI and MSS subtypes and comprehensively analyzing the unique immunometabolic characteristics of each subtype and potential therapeutic targets in order to reverse immunotherapy resistance.

## 2. Efficacy of ICIs in MSS CRC

### 2.1. Neoadjuvant Therapy

#### 2.1.1. Early-Stage CRC

The phase III study FOxTROT found that neoadjuvant chemotherapy (NAC) in patients with MSS CRC comprising 6 weeks FOLFOX+/−panitumumab achieved only a 3% pathological complete response (pCR) and an 8% major pathological response (MPR) (<10% residual tumor), corresponding to Mandard tumor regression grade 1 (complete response (CR)) or 2 (near CR) [10]. Only three studies (Table 1) have evaluated the efficacy of neoadjuvant immunotherapy (nIT) in MSS CRC. In a seminal paper, Chabali et al. [11] showed that 14% of their patients achieved a pCR and that 21% achieved an MPR with 6 weeks of nivolumab (two cycles) and ipilimumab (one cycle). In another study, 5% and 16% of patients treated with 6 weeks of nivolumab alone achieved a pCR and MPR, respectively [12]. Importantly, all patients underwent surgery after nIT, showing that this strategy is at least not detrimental to management. Another study with only Asian patients [13] found an impressive 51% pCR and 81.8% MPR with 12-week neoadjuvant therapy with FOLFOX + bevacizumab and sintilimab (six cycles). Because pCR and MPR rates of just 3% and 8% are expected with NAC (with the 6-week FOLFOX schedule without bevacizumab in the FOxTROT trial), this small study suggests a potential synergy among nIT, antiangiogenic therapy, and chemotherapy, at least in Asian patients.

#### 2.1.2. Early-Stage Rectal Cancer

Four phase II studies of ICI therapy in rectal cancer have been published as full papers (Table 2). All of these studies included RT, either short-course radiotherapy (SCRT) or long-course radiotherapy (LCRT), in combination with capecitabine. Two Asian studies with concomitant LCRT and ICIs (tislelizumab or nivolumab) showed pCR rates ranging from 30% to 50% [17,18]. Two other studies that evaluated the sequential use of SCRT followed by two cycles of CAPOX and camrelizumab [19] and the sequential use of LCRT followed by three cycles of durvalumab [20] achieved pCR rates between 32% and 48%.

Three randomized clinical trials (RCTs) have been published on ICI therapy in early-stage rectal cancer. One global study compared total neoadjuvant therapy (TNT) with induction FOLFOX followed by concomitant LCRT with or without pembrolizumab (six cycles) [14]. This study failed to meet the primary endpoint (increased pCR in the ICI arm). A trial including only Asian patients compared SCRT followed by CAPOX and camrelizumab (two cycles) vs. LCTR followed by CAPOX alone. The results indicated a significant increase in the pCR (39.8% vs. 15.3%) in the ICI arm [15]. Finally, one study evaluated, albeit as a secondary endpoint, if ICI therapy can improve the pCR rate. A small number of patients underwent nonoperative management (NOM) [16]. The study used a strategy of TNT with induction CAPOX with or without sintilimab (six cycles) followed by LTRT and surgery. There was a numerical increase in the pCR (36% vs. 24%) and NOM strategy (13.4% vs. 2.9%) in the sintilimab arm. Overall, the percentage of patients that underwent NOM was clearly inferior in both arms compared with the OPRA trial and it therefore seems that only a subset of patients desiring NOM was offered this strategy [21].

**Table 2 cancers-17-00498-t002:** Prospective randomized trials with ICI in MSS mCRC patients.

Trial	Number of Patients	Design/ Line (L)	Schedule	HR (90–95% CI)	mPFS/mOS (95% CI)	ESMO-MCBS PFS/OS
Tabernero et al., 2022 [22]	445	III/1L	FOLFOX followed by FL-B + atezolizumab vs. FL-B	0.92 (0.72–1.17)	7.1 (6.1–8.3) vs. 7.4 (5.9–9.1) *	1/1
Antoniotti et al., 2023 [23]	218	IIRz/1L	FOLFOXIRI-B + atezolizumab vs. FOLFOXIRI-B	0.77 (0.55–1.08) **	13.1 (12.5–13.8) vs. 11.5 (10–12.6)	2/1
Lenz et al., 2024 [24]	310	IIRz/1L	FOLFOX-B + nivolumab vs. FOLFOX-B	0.81 (0.53–1.23)	11.9 (8.9–15.7) vs. 11.9 (10.1–12.2)	2/1
Eng et al., 2019 [25]	363	III/>2L	Atezolizumab vs. atezolizumab + cobimetinib vs. regorafenib	1 (0.73–1.38)	7.1 (6.1–10.1) vs. 8.9 (7–10.6) vs. 8.5 (6.4–10.7)	1
Chen et al., 2020 [26]	180	IIRz/>2L	Durvalumab + tremelimumab vs. BSC	0.72 (0.54–0.97)	6.6 (6–7.4) vs. 4.1 (3.3–6)	1
Mettu et al., 2022 [27]	133	IIRz/>2L	Cape-B + atezolizumab vs. Cape-B	0.96 (0.63–1.45)	10.3 (8.3–15.2) vs. 10.2 (8.5–16.6)	1
RELATIVITY-123 (15 December 2023) [28]	700	III/>2L	Nivolumab + relatlimab vs. regorafenib or TAS102	NR	NR	1 ***
Kawazoe et al., LEAP-17, 2024 [29]	480	III/>2L	Pembrolizumab + lenvatinib vs. regorafenib or TAS102	0.83 (0.68–1.02)	9.8 (8.4–11.6) vs. 9.3 (8.2–10.9)	1
KEYFORM-007 (25 September 2024)	441	III/>2L	Pembrolizumab + favezelimab vs. regorafenib or TAS102	NR	NR	1 ***

Rz: randomized; NR: not reached; mOS, median overall survival; mPFS, median progression-free survival. * PFS from randomization after induction therapy to disease progression. ** CI in the MSS intention-to-treat population. *** Data have not been reported in International Meetings or been published.

### 2.2. Evaluating Clinical Benefit in Advanced CRC (ESMO-MCBS)

The original ESMO-MCBS (European Society for Medical Oncology-Magnitude of Clinical Benefit Scale) and the adapted version 1.1 [30,31,32] include an amendment that incorporates three evaluation forms for overall survival (OS): if the median OS with the standard treatment is ≤12 months, if the median OS with the standard treatment is >12 to ≤24 months, and if the median OS with the standard treatment is >24 months. There are two separate forms for progression-free survival (PFS): if the median PFS with standard treatment is ≤6 months and if the median PFS with standard treatment is >6 months. The ESMO-MCBS uses three parameters to qualify the benefit of new drugs in RCTs in each form. The lower limit of the 95% confidence interval (CI) of the corresponding hazard ratio (HR) and the median PFS or median OS are used to qualify the benefit in PFS and OS, respectively. The third parameter, the proportion of patients alive at a specific time (evaluated as a 10% difference), is used only to evaluate the benefit in OS. The lower limit of the 95% CI of the corresponding HR has been questioned because it is overly permissive. Fewer RCTs meet the higher ESMO-MCBS v1.1 scores when the estimated HR is used instead of the lower limit of the 95% CI. Variations in the width of the CI become narrower if the trial has mature data and a large sample size and wider if the trial has an immature follow-up and/or small sample size or the data are subjected to a subanalysis (typically in studies with pre-planned PD-L1 expression analysis) (Appendix A). We advocate for estimates instead of the lower limit of 95% of the CI to evaluate the ESMO-MCBS.

In advanced CRC, the median PFS usually ranges from 9 to 12 months, while the median OS ranges from 24 to 36 months. Therefore, the ESMO-MCBS 1.1 form scores used to evaluate the benefit in mCRC are a PFS >6 months and an OS >24 months. We have previously noted [33] that, for many reasons, PFS is a more vulnerable endpoint than OS. Therefore, a good correlation between PFS and OS is also desired in the ESMO-MCBS.

Several weaknesses with ESMO-MCBS version 1 should be highlighted. First, the lower limit of the 95% CI of the HR is very similar (0.65 and 0.70) in the PFS and OS forms. Despite the increase in the absolute number of months gained in median OS compared with median PFS, the median PFS estimate of the HR was 0.74 (range 0.64–0.84), while the median OS estimate of the HR was 0.85 (range 0.73–1.11) in seven RCTs in the first-line setting in esophageal/gastric cancer comparing ICI + chemotherapy vs. chemotherapy alone (Appendix A). If we use estimates, we advocate for the same HR stringency for PFS (<0.65) but for a less strict criterion for OS (<0.75). The second challenging aspect is the median benefit in PFS and OS. More than a 3-month gain in the >6 months PFS form corresponds to grade 3 (strongest recommendation), which correlates with a >9-month benefit in the >24 months OS form for grade 4. None of the published RCTs without pre-planned subanalyses showed this correlation (the correlation between the median PFS and median OS is usually 1:1 or 1:1.5). Therefore, a more conservative gain in median OS (3–5 months) would probably be desirable for a correlation with PFS (increase in >3 months in the median PFS).

Finally, a 10% absolute benefit at 2, 3, and 5 years is used only in the OS form and not in the PFS form. Because this parameter measures long-term benefit and is critically important, we advocate for its inclusion in the PFS form as well. In addition, as the 2-year PFS rate correlates well with the 3-year OS rate, we believe that absolute differences (e.g., a 10% or 20% absolute benefit) should be evaluated not only in OS but also in PFS.

### 2.3. Advanced Disease

Due to the poor efficacy of anti-PD-1 therapy in MSS mCRC patients (0% response rate) [9], several strategies have been developed in RCTs. RCTs using anti-PD-1 or anti-PD-L1 in combination in both first-line and heavily pretreated (more than two lines of therapy) MSS advanced CRC patients are analyzed in Table 2. Three trials in the first-line setting and six trials in heavily pretreated patients fulfilled the criteria and were analyzed according to ESMO-MCBS 1.1 criteria. We also include two RCTs that have not been published or presented in international meetings but that reported negative results.

In the first-line setting, the studies compared the addition of anti-PD-1 (atezolizumab in two studies [22,34] and nivolumab in one study [24]) to standard therapies (FOLFOX or FOLFOXIRI) + bevacizumab. Tabernero et al. added atezolizumab to FU/LV + bevacizumab after a common induction strategy with FOLFOX + bevacizumab (six cycles) while Antoniotti et al. used initial atezolizumab with FOLFOXIRI + bevacizumab. Neither of these two trials fulfilled the ESMO-MCBS 1.1 criteria. The third study used FOLFOX-bevacizumab with and without nivolumab and obtained negative outcomes.

Six RCTs have been completed in heavily pretreated patients. These RCTs used different strategies based on the synergism with anti-PD-1 identified in preclinical murine models of CRC [35,36]. In addition, phase II clinical trials suggest that these combinations increase anti-PD-1 efficacy, with an overall response rate ranging from 8% to 22%, comparing well with historical data using anti-PD-1 alone [37,38,39]. Despite promising clinical data from preclinical and phase II trials, RCTs with a MEK inhibitor (cobimetinib) [25], antiangiogenics (bevacizumab and lenvatinib) [27,29], and the ICIs anti-CTLA4 (tremelimumab) [26] and anti-LAG3 (relatlimab and favezelimab) [28,40] failed to increase the OS benefit compared with the standard of care. Overall, these RCT results questioned the value of current preclinical CRC murine models and the endpoints (e.g., overall response rate) that are currently used in phase II trials for adequate phase III trial development.

## 3. Potential Biomarkers of ICI Efficacy in MSS CRC

In the context of studies of biomarkers for assessing the efficacy of ICI therapy in MSS CRC, the application of REMARK (REporting recommendations for Tumor MARKer prognostic studies) criteria is essential to ensure that published scientific evidence meets high standards of transparency and methodological rigor. The REMARK guidelines were developed to improve the quality and consistency of tumor biomarker studies, particularly those with significant clinical implications. Thus, an evaluation under REMARK criteria involves prioritizing studies conducted in human patients rather than preclinical research, with a focus on prospective randomized trials—both in localized and metastatic disease—that provide robust and reliable data on the analytical validity and clinical utility of biomarkers. This approach not only enables clear interpretation of the evidence but also facilitates uniform comparisons across studies, allowing for the identification of potential biomarkers that may effectively guide personalized ICI therapies in patients with MSS CRC [41].

We propose an adaptation of the REMARK criteria for the MSS CRC population comprising three categories (A, B, and C). Category A would indicate a strong recommendation for biomarker use based on a study with a prospective design (including a prospectively calculated sample size for the biomarker, with predefined differences in HRs for PFS and/or OS, adjusted for other variables in advanced disease, or predefined differences in the pCR in the neoadjuvant setting); it must include a control arm (exposure to an alternative therapy to assess if the biomarker effect is prognostic or predictive) and incorporate a validation set (the use of at least one additional set with the same biomarker and cut-off). Category B would correspond to a non-prospective study that includes both a control arm and a validation set. Finally, Category C would include non-prospective studies lacking both a control arm and a validation set.

In Table 3, we summarize the biomarkers explored in immunotherapy studies in patients with MSS CRC. Various biomarkers have been examined in different studies, with many investigations focusing not on a single biomarker but on multiple biomarkers or their combinations. This approach likely stems from the observation that PD-L1, commonly used in other, more immunogenic, tumors, has shown limited utility in MSS CRC. An illustrative example of this combinatory strategy is the MEDITREM trial, which evaluated the safety and efficacy of combining durvalumab and tremelimumab with mFOLFOX6 chemotherapy as a first-line treatment for patients with RAS-mutant unresectable mCRC [42]. In that study, the tested biomarkers, such as a high tumor mutational burden (TMB), high intratumoral infiltration of CD8^+^ T cells, and low expression of decorin (a key component of the extracellular matrix that is notably upregulated in inflammatory cancer-associated fibroblasts (CAFs)), were associated with a better PFS [42]. Additionally, the study reported that immunomonitoring revealed the induction of neoantigen-specific, NY-ESO1-specific, and TERT-specific T cell responses in the blood, which were associated with improved PFS. Another notable proposal is the DetermaIO immune-related 27-gene expression signature, as demonstrated in the AtezoTRIBE study [23], where atezolizumab was added to first-line FOLFOXIRI (5-fluorouracil, oxaliplatin, irinotecan) + bevacizumab. DetermaIO holds potential as a predictive tool for identifying patients who may benefit from the addition of atezolizumab to first-line FOLFOXIRI + bevacizumab in advanced CRC.

It is worth noting that only one study used the consensus molecular subtype (CMS) classification as a biomarker. Their results showed that patients in the CMS3 group, known as the metabolic group, had better outcomes [24]. None of the studies found that TMB was associated with outcomes [24,43,44]. Finally, not only were T cell levels studied, but also those of other immune system cells, yielding interesting results. Parikh et al. investigated the addition of RT to an ipilimumab + nivolumab combination in patients with metastatic MSS CRC and advanced pancreatic adenocarcinoma. Their work highlighted the relevance of CAFs and natural killer (NK) cells in the treatment response of patients with metastatic disease. The expression of epithelial–mesenchymal transition (EMT)-related genes, enriched in responders, appears to be driven by CAF subtypes, particularly myofibroblastic CAFs, which are associated with better disease control in pretreatment biopsies. Additionally, responders exhibited significantly higher levels of resting NK cells, suggesting their role as key immune biomarkers for predicting the efficacy of combined ICI therapy and RT [44].

Finally, the HCRN GI14-186 phase 1b trial, which combined pembrolizumab with chemotherapy in mCRC patients, identified a wide variety of both soluble and cellular biomarkers associated with the clinical response and PFS [45]. The soluble biomarkers included baseline levels of TNF-α (lower levels were associated with a better RECIST response), Flt3 ligand, and TGF-α (higher levels were linked to improved PFS), while increased CCL5 during treatment correlated with worse outcomes. Immune checkpoints such as LAG3 (lower baseline expression on CD4^+^ and CD8^+^ T cells was linked to better outcomes), BTLA, and VISTA also showed potential, suggesting the need for combined checkpoint inhibition. Additionally, low baseline levels of CD8 ^+^ FasR^+^ T cells expressing PD-1 and high levels of Tc17 (CD8^+^RORγt^+^PD-1^+^) T cells correlated with better outcomes, highlighting the roles of apoptosis and T cell plasticity in the treatment response [45].

## 4. Role of Metabolism in ICI Resistance

The introduction of immunotherapy with ICI has revolutionized cancer treatment across various tumor types, establishing itself as a cornerstone in many therapeutic regimens. This approach has significantly improved response rates and survival outcomes in several cancers. Nevertheless, its effectiveness is inconsistent; while many patients benefit, others do not respond, and some even experience rapid disease progression [46]. The reasons for this variability remain unclear, and it is still not fully understood why ICIs are ineffective in certain tumor types. For this reason, a wide spectrum of tumor biomarkers is being studied to enhance the outcomes of ICI therapy. One approach, which represents a hallmark of cancer research, is to understand the role of metabolism in the immune response and its influence on the tumor microenvironment (TME) [47]. Metabolism and immune cell behavior are closely intertwined, and their relationship influences both immune evasion mechanisms and nutrient competition between tumor and immune cells [48]. The metabolic demands of immune cells are directly linked to the success of immunotherapy. Therefore, improved understanding of the metabolic differences between long-term responders and rapid progressors could be instrumental in identifying the most effective treatment strategies [48].

Several metabolic mechanisms could be implicated in resistance to ICI therapy. A better understanding of these mechanisms and their role within the TME is crucial to elucidating the underlying causes of this resistance and, ultimately, developing more effective therapeutic strategies. In the following section, we delve into the key metabolic mechanisms identified as contributors to ICI resistance. These mechanisms, which operate within the TME, are essential for understanding how tumors evade immune responses and why some patients fail to respond to treatment.

First, oxidative phosphorylation (OXPHOS), a mitochondrial pathway for energy production, is critical in certain cancer subtypes. While many tumors predominantly rely on aerobic glycolysis, a phenomenon known as the Warburg effect, some tumors shift to OXPHOS to adapt to low-glucose conditions. This metabolic reliance not only enhances energy efficiency but also supports resistance mechanisms by fostering an immunosuppressive TME [49]. For instance, in melanoma brain metastases, RNA sequencing reveals a significant enrichment of OXPHOS pathways compared to extracranial disease, which correlate with poor prognosis and a reduced response to MAPK-targeted therapies [50]. Furthermore, ICI-resistant melanoma cells exhibit a hypermetabolic state by upregulating both glycolysis and OXPHOS. This dual adaptation intensifies oxidative stress and promotes a hypoxic environment through overexpression of enzymes such as ADH7, limiting the infiltration and function of cytotoxic CD8^+^ T cells and thereby enhancing immune resistance [51].

In contrast, glycolysis represents another key metabolic adaptation in tumors, enabling rapid proliferation and immune evasion. This process, consisting of ten enzymatic reactions, converts glucose into pyruvate to generate energy and produce essential intermediates for biosynthesis. In many cancers, glycolysis is preferred over OXPHOS, even in the presence of oxygen, due to the Warburg effect [52]. This metabolic shift supports rapid proliferation by providing both adenosine triphosphate (ATP) and the intermediates necessary for lipid, nucleotide, and amino acid synthesis. Additionally, glycolysis produces lactate, a metabolite that contributes to immune evasion and facilitates the development of a supportive TME, enabling sustained tumor growth. This reliance on glycolysis underscores its importance in driving aggressive tumor behavior and resistance to therapies, including immunotherapy. Monocarboxylate transporter 4 (MCT4) plays a crucial role in the tumor glycolysis pathway, particularly in promoting resistance to immunotherapy [53]. For example, glycolysis is critically involved in lung adenocarcinomas (LUADs) with *STK11/LKB1* mutations, key genomic drivers of primary resistance in *KRAS*-mutated tumors. The loss of *LKB1* enhances lactate production and secretion via MCT4, promoting immune evasion. Single-cell RNA profiling has shown that *LKB1*-deficient tumors exhibit increased M2 macrophage polarization and dysfunctional T cells, phenomena that can be replicated by exposure to exogenous lactate and mitigated by MCT4 knockdown or blockade of the lactate receptor GPR81 on immune cells. Notably, genetic ablation of MCT4 reverses the PD-1 resistance caused by LKB1 loss in murine models, highlighting the role of lactate-driven immunosuppression. Similarly, *STK11/LKB1*-mutant LUAD tumors in patients demonstrate enhanced M2 macrophage polarization and T cell dysfunction, emphasizing the therapeutic potential of targeting MCT4 and the lactate pathway to overcome immunotherapy resistance in glycolytic tumors [54].

Glycosylation is a critical post-translational modification whereby carbohydrates are attached to proteins, notably through N-linked glycosylation, which is regulated by enzymes such as glycosyltransferases. This modification affects protein structure, stability, and interactions and plays vital roles in various biological processes, including immune regulation. For example, glycosylation of T cell receptors can influence immune responses, although the specific impact on proteins such as PD-1 remains under investigation [55,56]. For instance, glycosylation plays a significant role in triple-negative breast cancer by modulating immune checkpoint interactions. Specifically, epidermal growth factor induces PD-L1 and PD-1 binding through the enzyme B3GNT3, which is essential for immune evasion. A reduction in B3GNT3 levels has been shown to boost T cell-mediated antitumor immunity. Additionally, a monoclonal antibody targeting glycosylated PD-L1 not only blocks the PD-L1/PD-1 interaction but also promotes PD-L1 degradation and induces direct cancer cell killing, suggesting that glycosylation targeting may enhance immune checkpoint therapy in breast cancer [57].

Finally, gluconeogenesis adds another layer of complexity to the metabolic landscape in tumors, particularly those adapting to nutrient scarcity. This energy-intensive process synthesizes glucose from precursors such as lactate, amino acids, and glycerol, providing essential resources for survival in low-glucose environments. Enzymes such as glucose-6-phosphatase and phosphoenolpyruvate carboxykinase regulate gluconeogenesis, ensuring metabolic flexibility. In tumors, this pathway supports biosynthesis and energy homeostasis, indirectly influencing immune cell function within the TME [58].

In summary, metabolic adaptations within the TME, including OXPHOS, glycolysis, glycosylation, and gluconeogenesis, play pivotal roles in driving resistance to ICI therapy. Improved understanding of these mechanisms would offer critical insights into immune evasion strategies and highlight potential therapeutic targets to enhance the efficacy of immunotherapy.

## 5. Impact of Immunotherapy in CRC

As previously noted, CRC, unlike other cancers, does not represent a clear case in which therapies based on PD-1 or PD-L1 inhibition have transformed treatment paradigms or significantly altered disease progression over the years. Currently, polychemotherapy regimens incorporating fluoropyrimidines and platinum-based agents, in combination with therapies targeting either *EGFR* or VEGF, remain the standard approach, emphasizing the fact that CRC is a histological subtype with limited responsiveness to ICI therapy.

Nonetheless, MSI/MMRd tumors exhibit markedly distinct behavior from MSS/MMRp tumors in terms of the immunotherapy response. In both disseminated and localized disease, MSI/MMRd patients demonstrate significantly higher response rates and improved OS [59]. In localized disease, a 100% pCR and a 40% reduction in recurrence have been reported in patients treated with anti-CTLA4 and anti-PD-1 combinations [11].

Differences in immunological and metabolic features between MSS and MSI tumors could explain the disparity in their responses to immunotherapy. A better understanding of these differences could help us to refine therapeutic strategies and reduce acquired resistance.

### 5.1. Non-Metabolic Differences Between CRC MSS/MSI: The TME

The TME comprises multiple entities that create a dynamic and highly variable environment. A better understanding of the differences between MSS and MSI CRC tumors can provide insights into their distinct behavior in response to the above-mentioned ICI therapies. However, it is well established that a high lymphocytic infiltration within CRC tumors, particularly with a predominance of T helper 1 (Th1) cells characterized by IFN-γ expression, is associated with a favorable prognosis due to their potentiation of the immune system. In contrast, profiles dominated by T helper 17 (Th17) cells and linked to IL-17 expression tend to correspond to worse outcomes because of their role in promoting tumor growth [60,61]. These findings are predominantly observed in tumors with MSI, which, by definition, exhibit a significantly higher TMB—10 to 50 times greater—compared with MSS tumors. This elevated mutational load leads to the formation of more neoantigens and neoepitopes, the segments recognized by T cells, thereby enhancing immune recognition and response efficacy [62,63].

Regarding ICI expression, MSI tumors typically show higher levels of PD-1, PD-L1, LAG-3, and indoleamine 2,3-dioxygenase (IDO), an enzyme that depletes tryptophan and inhibits T cell proliferation in infiltrating cells [64], particularly within myeloid cells of the TME, indicating potential key points of immune evasion. Moreover, the expression of proinflammatory cytokines associated with Th1 responses, such as IFN-γ and TBX21 (the gene encoding T-bet), is notably lower in MSS tumors. There are substantial differences between MSS and MSI CRC tumors across the three primary compartments of the TME: the tumor-infiltrating lymphocyte (TIL) zone, the stromal region, and the invasive front (where the tumor invades the lamina propria). MSI tumors are characterized by a higher density of infiltrating cells within the TIL zone and stroma, with a significant increase in CD8^+^ T cells. Furthermore, MSI tumors tend to exhibit increased FOXP3 expression in the TME, corresponding to a higher infiltration of regulatory T cells (Tregs) compared with MSS tumors. Although it may seem counterintuitive that MSI tumors contain more Tregs, it is reasonable to consider that a heavily infiltrated tumor requires a substantial presence of Tregs to maintain the immune balance. Therefore, it is not a matter of quantity but of proportion with respect to effector T cells [60]. This Treg infiltration may serve to modulate the adaptive immune response against an aggressive and proinflammatory TME, thereby containing excessive immune activation [65,66].

Unlike other tumors, CRC appears to show substantial heterogeneity in myeloid cell populations within its TME. Multiple tumor-associated macrophage (TAM) populations have been identified, likely derived from distinct differentiation processes, along with considerable variability among dendritic cell (DC) populations [67]. This suggests that not all TAMs are immunosuppressive; some may have proinflammatory functions, and it appears that this activity could potentially be defined by various features of the TME [68]. These TAMs may serve as central nodes within the CRC network. This distinct variability differentiates the TME of CRC from that of other tumor types, which tend to show greater continuity and homogeneity among cell populations [69,70]. In addition, differences in myeloid cell populations have been observed between MSS and MSI tumors. Transcriptomic studies of MSI tumors with advanced sequencing, spatial, and gene expression analyses have detected hubs in the stromal zone, at the tumor–luminal interface of myeloid cells (macrophages, DCs, and granulocytes) and CXCR3-ligand+ effector T cells [71,72] Therefore, the spatial localization of these cell groups is not random: they are strategically placed to coordinate with T cells to interact with tumor cells.

In MSS tumors, T cell–hub interactions appear to be limited due to inadequate signaling compared with MSI tumors. This limitation is due to differences in chemokines, cytokines, and adhesion molecules and the unique composition of the TME in MSS cancer, such as the increased presence of TGF-β and immunosuppressive behavior of stromal cells, including fibroblasts [72]. In MSS tumors, at least in the mesenchymal subtype, fibroblasts exhibit a notably more active proinflammatory profile, leading to a TME that supports cancer progression [73]. These fibroblasts, often referred to as inflammatory CAFs (iCAFs), secrete a variety of growth factors and chemokines, such as CXCL8, along with matrix metalloproteinases (including MMP2 and MMP3). This secretion not only promotes tissue remodeling but also the angiogenesis process and recruits immune cells such as neutrophils and T lymphocytes to the tumor site [74]. This interaction fosters an inflammatory environment that may enhance the elimination of tumor cells but also has the potential to contribute to therapy resistance and tumor progression.

Nevertheless, the behavior of CAFs in CRC appears to be quite plastic and malleable according to the circumstances of the tumor. On the luminal surface of colonic tumors, an inflammatory core characterized by positive feedback among iCAFs, monocytes, and neutrophils is established, which promotes inflammation and remodeling of the tumor tissue. This region, due to its exposure to abrasive damage by intestinal contents, facilitates the entry of microbes and immunostimulatory molecules, triggering an inflammatory response associated with tissue repair. Moreover, IL-1-induced inflammation can activate RSPO3 expression in mesenchymal cells and, together with IL-17 activity and matrix metalloproteinases (e.g., MMP2 and MMP3) produced by iCAFs, stimulates angiogenesis and promotes tumor progression [74,75,76]. In this context, the reduction observed in MSI tumors of CAFs expressing CXCL14 and bone morphogenetic proteins (BMPs) is relevant, in contrast to MSS tumors where these CAFs are more abundant. These CAFs normally inhibit the Wnt pathway, a key signal for proliferation and metastasis development, thereby promoting epithelial differentiation. It therefore appears that CAFs in MSI cancer do not help to promote the proinflammatory state of the TME [77] and may even be part of a tumor evasion mechanism and a resistance mechanism to immunotherapy-based therapies [72]. CAFs also contribute to the expression of stem cell niche factors, such as RSPO3 and GREM1, which are widely expressed in tumors, unlike their restricted expression in normal tissue. Their high expression drives tumor growth [78], but it appears that their distribution is important. In MSI cancer, these proteins are abnormally redistributed in the TME, arranged in stromal bands extending upward from the tumor base, which may promote tumor growth and aggressiveness. In contrast, in MSS cancer, their expression is limited to specific areas of normal tissue, suggesting a tighter control that may be associated with less aggressive tumor behavior [72]. This implies that CAFs can change their function depending on the TME where they are located.

To conclude, the interplay among immune cells, CAFs, and myeloid populations within the TME highlights critical differences between MSI and MSS CRC tumors. These variations in immune infiltration, fibroblast activity, and cytokine signaling underpin distinct mechanisms of immune evasion and therapy resistance. Improved understanding of these dynamics would provide valuable insights for tailoring immunotherapeutic strategies and targeting specific vulnerabilities in CRC subtypes.

### 5.2. Metabolic Competition in the TME

In general, most cells generate energy to maintain cellular processes through mitochondrial OXPHOS, producing ATP. However, in the absence of oxygen, they employ a less efficient metabolic pathway: glycolysis [79]. Interestingly, cancer cells use glycolysis even in the presence of oxygen, a phenomenon known as aerobic glycolysis (also called, as noted above, the Warburg effect) [80]. Figure 1 provides an overview of the key interactions between metabolism and the immune system, summarizing their essential connections.

Nutritional competition, defined as the demand for glucose or glutamine by tumor cells and T cells, is physically established in the TME. The high demand for glucose by tumor cells, together with the limited availability of this nutrient in CD8^+^ T cells, may compromise their function and activity. In addition, PD-L1 expression in tumor cells activates the Akt/mTOR pathway, further promoting glycolysis [80]. On the other hand, CD4^+^ T cells are also affected, resulting in increased production of immunosuppressive factors, such as TGF-β [81]. This competitive and metabolically restricted environment contributes significantly to immunosuppression in the TME and represents a critical challenge for immune therapies. Anti-PD-L1 antibody therapy can reverse these effects by blocking PD-L1 signaling, which increases extracellular glucose availability and improves the function of CD8^+^ and CD4^+^ T cells and reduces their exhaustion [79]. This approach is particularly relevant in T cell-infiltrated tumors, such as MSI/MMRd tumors.

#### 5.2.1. Metabolic Mechanisms Shaping Immune Responses

In addition to the relationship between PD-L1 and glucose competition, a direct connection has been established between glucose availability and T cell function. Insufficiency of the glycolytic metabolite phosphoenolpyruvate (PEP) impairs Ca^2+^-NFAT signaling and compromises T cell activation due to an increase in sarcoplasmic/endoplasmic reticulum Ca^2+^-ATPase (SERCA)-mediated Ca^2+^ reuptake. This implies that, under low-glucose conditions, reduced PEP production negatively impacts the ability of T cells to activate and respond effectively. Studies in murine models have shown that metabolic reprogramming of T cells to increase PEP production could be a promising strategy to strengthen antitumor immune responses. This approach could also enhance the efficacy of therapies based on adoptive T cell transfer (e.g., CAR-T) [81,82].

Another crucial aspect in the metabolic competition between tumor cells and T cells lies in the demand for glutamine. This amino acid is essential for T cells, as it promotes the production of acetyl-CoA at the mitochondrial level, facilitating histone acetylation and chromatin opening in genes associated with immune memory. This underlines the importance of glutamine in the differentiation of T cells toward a long-lasting memory phenotype. In murine models, short-term inhibition of mitochondrial pyruvate carrier (MPC) in a CAR-T model has been shown to enhance glutamine utilization for acetyl-CoA production and fatty acid oxidation, thereby favoring differentiation toward a memory phenotype in T cells [83]. Additionally, isocitrate dehydrogenase 2 (IDH2) plays a key role in the reductive carboxylation of glutamine in CD8^+^ effector T cells. Although IDH2 is not essential for the proliferation or effector function of these cells, its inhibition alters metabolic pathways, leading to a metabolite imbalance that appears to favor differentiation into memory T cells [84].

In a TME rich in cytotoxic T cells, lactate plays an important role as an energy source for these cells. The oxidation of lactate occurs through its conversion to pyruvate by lactate dehydrogenase (LDH). Subsequently, pyruvate is transported to mitochondria via MPC, where it is metabolized in the citric acid cycle to generate ATP [85]. Within mitochondria, pyruvate is converted to acetyl-CoA, a key metabolite in energetic and biosynthetic processes. Furthermore, it has been observed that temporary or short-term inhibition of MPC can activate alternative metabolic pathways, such as that mediated by pyruvate dehydrogenase (PDH), to convert pyruvate to acetyl-CoA. This increase in acetyl-CoA levels has significant implications for histone acetylation and epigenetic regulation. Such epigenetic changes may influence T cell differentiation and function, enhancing their responsiveness to tumor cells [83]. These findings highlight the relevance of metabolic mechanisms in immune function and their potential as a therapeutic target to optimize antitumor responses.

#### 5.2.2. Therapeutic Implications in MSI and MSS CRC

The functional exhaustion of cytotoxic T cells represents a major challenge in the field of immunotherapy. Exhausted T cells do not respond effectively to ICIs, which has led to the investigation of reactivation strategies. Treatment with an IL-10 fusion protein (IL-10-Fc) has been evaluated in murine models of colorectal (CT26) and melanoma (YUMM1.7-OVA) tumors. This treatment promotes OXPHOS in exhausted T cells, with the aim of restoring their activity against tumor antigens. Treated mice showed significantly higher survival rates than untreated controls. Furthermore, CT26 model mice that survived the initial treatment demonstrated an enhanced ability to reject a new tumor cell implant, suggesting effective immunological memory [86]. These findings highlight the potential of metabolic and functional reprogramming approaches to overcome T cell exhaustion and improve the efficacy of immune therapies.

Although it may seem that nutritional competition in the TME occurs exclusively between immune and tumor cells, the balance of the TME is influenced by a variety of entities that can dynamically change their functionality depending on the environmental conditions. Among the most glucose-consuming entities are myeloid cells [87,88], specifically TAMs with immunosuppressive activity. TAMs respond to environmental signals such as hypoxia and proinflammatory cytokines, which can induce metabolic adaptations that enhance glucose uptake and activate metabolic pathways, allowing survival and functionality under adverse conditions. A key regulatory mechanism in glucose uptake by myeloid cells involves the mammalian target of rapamycin complex 1 (mTORC1) signaling pathway, which governs cell growth and metabolism. mTORC1 activation promotes the expression of glucose transporters such as SLC2A1 (GLUT1) and enzymes such as hexokinase (HK2 and HK3), facilitating glucose utilization and nutrient acquisition in the TME [88].

Traditionally, glucose has been viewed as a limiting factor in the TME. However, emerging evidence suggests that immunosuppression in the TME is not solely due to glucose scarcity but rather the metabolic allocation and preferences of specific cell subtypes. For instance, blockade of glutamine metabolism in the TME has been shown to increase glucose consumption in both immune and tumor cells [88]. Selective inhibition of glutaminase via CB-839 blocks the conversion of glutamine into glutamate, leading to enhanced glycolysis and lactate production and compromised OXPHOS. This allows Tregs to maintain their suppressive function while promoting metabolic adaptation, although with reduced IFN-γ production, in Th1 cells. In contrast, pan-glutamine inhibition using 6-diazo-5-oxo-L-norleucine (DON) disrupts multiple glutamine-dependent pathways, significantly impairing Treg proliferation and function, diminishing Th1 responses to intracellular pathogens, and reducing Th17 differentiation. Total glutamine depletion drives metabolic adaptations that sustain cell survival but limit T cell functionality and antigen responsiveness, emphasizing the central role of glutamine in immune regulation [89,90].

Effector T cells demonstrate superior metabolic plasticity to the rigid and less adaptable cancer cells. Upon glutamine inhibition, cancer cells exhibit reduced ATP production and biosynthesis of essential components (e.g., lipids and proteins), leading to tumor growth stagnation, hypoxia, and acidosis. These conditions create a hostile microenvironment that further impairs immune cells. In contrast, effector T cells can adapt by enhancing oxidative metabolism to efficiently generate ATP via cellular respiration, acquiring long-lived phenotypes [91]. This metabolic resilience enables T cells to function effectively in the hostile TME while cancer cells weaken. For example, murine TNBC tumor models showed that the blockade of glutamine metabolism in tumor cells leads to increased glutathione synthesis in CD8^+^ T cells, enhancing their oxidative stress resistance and functionality [92]. Similarly, in murine urological tumor models, the glutamine antagonist JHU083 reprograms TAMs by increasing glycolysis, disrupting the tricarboxylic acid (TCA) cycle and purine metabolism, enhancing tumor cell phagocytosis, and reducing proangiogenic capacity. JHU083 also suppresses glutamine-dependent metabolic pathways in tumor cells, decreasing HIF-1α and c-MYC expression while promoting apoptosis [93]. However, glutaminase inhibition in specific tumor contexts, such as in STK11/LKB1-deficient lung cancer, may impair CD8^+^ T cell activation by limiting the expansion of T cell receptor clonotypes, a critical component for robust antitumor immunity [94]. Moreover, activated CD8^+^ T cells are highly dependent on glutamine availability and glutaminase activity for their metabolic demands, these elements being necessary for effective proliferation and cytokine production. Therefore, while the targeting of glutamine metabolism represents a promising strategy to weaken cancer cells, it must be carefully balanced to avoid compromising the immune response necessary for effective immunotherapy.

In the context of CRC, particularly in MSI tumors, these findings are highly relevant. MSI tumors, characterized by high mutational burden and immune infiltration, are more responsive to immune-modulating strategies, including those targeting metabolic pathways such as glucose and glutamine metabolism. By weakening tumor cells and enhancing the metabolic adaptability of immune cells, these strategies may potentiate antitumor responses. These effects are especially promising in MSI CRC, where the immunogenic nature of the tumors aligns well with the metabolic and immune reprogramming induced by glucose and glutamine inhibition; however, the data are also important for understanding and ultimately exploiting the metabolic mechanisms underlying the conversion of MSS tumors into more proinflammatory MSS tumors.

### 5.3. Metabolic Pathways in MSS CRC

The metabolic adaptations of CRC vary significantly among its molecular subtypes, influencing both tumor growth and response to therapy. In MSS CRC subtypes, two dominant patterns emerge: the mesenchymal subtype, characterized by glycolysis-driven pathways, and the epithelial subtype, which displays a more heterogeneous reliance on OXPHOS, gluconeogenesis, and glycolysis. These distinct metabolic features not only fuel tumor progression but also shape the immunosuppressive TME, affecting immune surveillance and therapy resistance. An improved understanding of these metabolic landscapes would offer critical insights into the vulnerabilities of MSS CRC tumors and shed light on potential therapeutic strategies tailored to their metabolic dependencies. In the following sections, we delve into the metabolic hallmarks of the mesenchymal and epithelial MSS CRC subtypes, emphasizing their respective interactions with the TME, immune cell dynamics, and therapeutic implications. In Figure 1, we provide an overview of the key interactions between metabolism and the immune system across the different subtypes.

#### 5.3.1. Metabolic Patterns in Mesenchymal MSS CRC Tumors

The mesenchymal MSS CRC subtype of tumors exhibits distinct metabolic adaptations that correlate with an aggressive and treatment-resistant phenotype reminiscent of the EMT. These tumors have adapted to a hostile TME characterized by acidity, hypoxia, and limited nutrient availability, conditions that reinforce their invasive and resilient nature. A defining feature of this mesenchymal group is a marked upregulation of glucose transporters, glycolytic enzymes, and lactate transporters (e.g., SLC16A1, which also transports branched-chain ketoacids), pointing to a high dependency on glucose and glycolysis as primary sources of carbon, energy, and metabolic intermediates [95]. As previously mentioned, this reliance on glycolysis over oxidative metabolism is coupled with the reduced expression of key mitochondrial components necessary for pyruvate transport (e.g., MPC1) and oxidation (e.g., isocitrate dehydrogenase isoenzymes and mitochondrial respiratory complexes). Consequently, these tumors exhibit minimal OXPHOS and TCA cycle activity, which aids their survival under low-oxygen conditions. Additionally, the overproduction of immunosuppressive metabolites such as kynurenine contributes to immune evasion, further promoting the invasive and sustained growth typical of the mesenchymal phenotype [95].

Therefore, the glycolysis mechanism plays an essential role in the mesenchymal subgroup. Tumor-derived lactic acid suppresses T and NK cell function, which contributes to immune evasion. Glycolysis targeting may offer a therapeutic advantage in high-glycolytic tumors resistant to immunotherapy. Diclofenac, a non-steroidal anti-inflammatory drug (NSAID), inhibits lactate transporters (MCT1/4), reducing lactate secretion and enhancing anti-PD-1 efficacy. Unlike other NSAIDs, diclofenac preserves T cell activation and function under low-glucose conditions and delays tumor growth in vivo, highlighting its potential as a complementary strategy to ICI therapy in high-glycolytic cancers [96]. As mentioned above, MCT4 targeting reduces lactate efflux and reverses lactic acid-driven immunosuppression. In 3D CRC spheroid models, MCT4 inhibition enhances T cell function, immune cell infiltration, and spheroid viability. Combined with ICI therapy, it increases intratumoral pH, delays tumor growth, and prolongs survival in vivo. Notably, MCT1 inhibition offers no additional benefit, highlighting the role of MCT4 as a promising therapeutic target to improve ICI efficacy in glycolytic tumors [53]. The mesenchymal subtype is also defined by the accumulation of hyaluronan and reduced levels of atypical protein kinase C, both of which are linked to poor clinical outcomes. Hyaluronan promotes epithelial heterogeneity and supports the emergence of tumor fetal metaplastic cells with invasive characteristics through interactions with activated fibroblasts. In addition, hyaluronan targeting using hyaluronidase has demonstrated efficacy in reducing tumor growth, limiting liver metastasis, and improving the effectiveness of ICI therapy by enhancing the infiltration of B and CD8^+^ T cells, including subsets with resident memory phenotypes, while mitigating immunosuppressive mechanisms [97].

Additionally, the high glycolytic activity in mesenchymal tumors supports increased glycosylation, a post-translational modification that aids immune evasion and enhances cell adhesion, promoting invasion and metastasis. This over-glycosylation not only affects PD-L1 on tumor cells but also impacts PD-1 on T cells, where N-glycosylation is crucial for maintaining protein stability and mediating interactions with PD-L1, particularly at the N58 site. Recent work has shown that the targeting of glycosylated PD-1 with high-affinity monoclonal antibodies disrupts PD-L1/PD-1 binding more effectively than conventional antibodies, enhancing antitumor immunity [55]. These insights highlight glycosylation as a crucial factor in immune evasion and suggest that the targeting of glycosylated PD-1 and PD-L1 could improve immunotherapy outcomes [98].

To further explore how mesenchymal tumor cells interact with the TME, we must examine how their unique metabolic characteristics impact both T cells and myeloid cells. First, high levels of LDH-A in tumors, which promote lactic acid production, are associated with poor cancer prognosis. Elevated lactic acid impairs T and NK cell activation through NFAT inhibition, thereby reducing IFN-γ production and facilitating immune evasion. Tumors with lower lactic acid production demonstrate slower growth and increased infiltration of IFN-γ-producing T and NK cells [99]. The immunosuppressive role of lactic acid in the TME is strengthened by the unique metabolic adaptations of Tregs, which are regulated by the transcription factor Foxp3. Foxp3 enables Tregs to limit Myc signaling and glycolysis while promoting OXPHOS and NAD regeneration, allowing them to thrive in high-lactate, low-glucose conditions. This metabolic flexibility helps Tregs to resist the NAD-depleting effects of L-lactate that impair effector T cells, thereby promoting immune tolerance while limiting antitumor immunity in the TME [100]. In highly glycolytic tumors with low glucose availability, Treg cells absorb lactic acid via the MCT1 transporter, which increases PD-1 expression and reinforces their suppressive function. This upregulation of PD-1 in Tregs, compared to effector T cells, can limit the efficacy of PD-1 blockade therapies, indicating that lactic acid acts as a regulatory checkpoint in the TME to enhance Treg-mediated immune suppression [101]. Interestingly, recent work revealed that, while lactate uptake is dispensable for Treg function in peripheral tissues, it becomes essential within tumors. Knockout of the lactate transporter MCT1 in Tregs slows tumor growth and enhances responses to immunotherapy, suggesting that Tregs in the TME rely on alternative metabolites to maintain their suppressive functions. This indicates a dual survival strategy is utilized by tumors that comprises depriving effector T cells of crucial nutrients while simultaneously supporting the metabolism of regulatory cell populations to promote immune evasion [102].

Moreover, immune cell function is suppressed in low-glucose, high-lactate environments, which notably impacts glycolysis-dependent T cells. Lactate disrupts T cell metabolism by reducing NAD^+^, blocking glycolytic enzymes (GAPDH, PGDH) and depleting the intermediates, such as serine, needed for T cell proliferation. Serine supplementation restores T cell function, suggesting that modulation of the redox balance may enhance cancer immunotherapy outcomes [103]. Another player in this field is MPC. This transporter regulates CD8^+^ T cell differentiation by promoting metabolic flexibility. MPC inhibition shifts T cells toward a memory phenotype, enhancing gene activation linked to longevity. However, in the TME, MPC is essential for lactate oxidation to support antitumor functions. The preconditioning of CAR-T cells with an MPC inhibitor primes them for lasting antitumor activity, underscoring the importance of mitochondrial pyruvate metabolism in enhancing immune responses against cancer [83]. Additionally, the availability of physiological carbon sources significantly influences glucose metabolism in CD8^+^ T cells. Rather than relying solely on glucose, CD8^+^ T cells can utilize lactate directly to fuel the TCA cycle. This lactate-based metabolism supports T cell homeostasis and proliferation. In the context of the TME, where lactate is abundant, CD8^+^ T cells may similarly depend on lactate as an alternative energy source, shaping their metabolic response and function within the tumor [104].

T cells are not the only affected cells in this subgroup with a mesenchymal metabolic pattern. Macrophages, the immune cells essential for tissue repair and defense, contribute to antitumor immunity by phagocytosing and eliminating cancer cells. However, in the TME, metabolic by-products from tumor cells, particularly those produced via glycolytic pathways, can drive pro-tumor signaling. These metabolites stimulate vascular endothelial growth factor (VEGF) expression and induce an M2-like polarization of TAMs through activation of hypoxia-inducible factor 1α (HIF1α). This, in turn, increases arginase 1 levels, creating a tumor-supportive environment. This interaction illustrates a communication mechanism between macrophages and tumor cells that originally evolved for tissue homeostasis but is repurposed by tumors to promote growth [105].

Therefore, macrophages assume a multifaceted role in tumor metabolism and immune resistance, particularly through the M2-polarized TAMs that suppress T cell migration and activation. In tumors undergoing EMT mediated by the Zeb1 transcription factor, TAMs become further polarized toward an M2 phenotype. This shift promotes immunotherapy resistance by creating an immunosuppressive microenvironment. These findings indicate that ZEB1 mediates dual resistance pathways by promoting PD-L1 and CD47 on invasive tumor cells, which induces M2 polarization in adjacent TAMs. This metabolic reprogramming shields tumor cells from the surrounding inflammatory signals, allowing them to evade immune responses effectively and sustain their growth [106]. Cortés et al. report that ZEB1 plays a critical role in macrophage plasticity by promoting their transition to an immunosuppressive state, which is essential for both initiating and resolving inflammation. They found that metformin, known for its anti-inflammatory and antioxidant effects, relies on ZEB1 expression to exert these actions in macrophages. By modulating mitochondrial function, through enhanced autophagy and reduced mitochondrial protein synthesis, ZEB1 facilitates the shift from inflammatory to immunosuppressive states. Metformin mimics this ZEB1-driven metabolic reprogramming in myeloid cells, underscoring the potential of ZEB1 targeting to modulate inflammation and immunosuppression within the TME [107]. Interestingly, macrophage metabolism can be reprogrammed to enhance the antitumor potential of macrophages. Activation with CpG oligodeoxynucleotide, a Toll-like receptor 9 agonist, reshapes macrophage metabolism, allowing macrophages to overcome tumor cell CD47-mediated inhibition. This metabolic shift relies on fatty acid oxidation and specific enzymes, positioning central carbon metabolism as a promising target to amplify macrophage antitumor activity in the TME [108].

#### 5.3.2. Metabolic Patterns in Epithelial MSS CRC Tumors

The epithelial MSS CRC subtype exhibits diverse metabolic adaptations, enabling tumor growth and immune evasion in a challenging TME. While some tumors prioritize alternative carbon sources such as glutamine, branched-chain amino acids (BCAAs), fatty acids, and acetate, others rely heavily on OXPHOS and glycolysis to sustain energy and biosynthesis. Enhanced gluconeogenesis pathways, supported by enzymes such as PCK2 and FBP1/FBP2, further underscore this metabolic plasticity. Upregulated fatty acid oxidation and acetyl-CoA synthesis allow tumor cells to survive in glucose-deprived environments. Autophagy and lysosomal activity also play key roles in immune evasion by degrading MHC-I molecules, which limits antigen presentation [95]. In contrast, other epithelial MSS CRC tumors depend heavily on OXPHOS and glycolysis, marked by upregulation of electron transport chain components, pentose phosphate pathway (PPP) enzymes, and glycolytic transporters such as GLUT1. This process supports the mitochondrial metabolism, energy production, and biosynthesis essential for rapid proliferation [95]. Moreover, this metabolic diversity highlights two distinct strategies: adaptation to glucose scarcity via gluconeogenesis and alternative carbon sources or reliance on OXPHOS and glycolysis for proliferative demands. These vulnerabilities suggest therapeutic avenues, including the targeting of autophagy, glutamine metabolism, or PPP enzymes, to counter tumor progression and immune resistance in epithelial MSS CRC tumors.

The OXPHOS metabolic pathway is closely associated with resistance to ICIs in MSS CRC patients due to its role in creating an immunosuppressive TME. Tumor-infiltrating T cells often display significant impairments in oxidative metabolism, such as defective mitochondrial biogenesis, fusion, and function, which contribute to T cell exhaustion independently of PD-1 signaling [109]. Furthermore, metabolic disruptions in tumor cells, such as hypoxia, not only enhance tumor progression but also correlate strongly with ICI resistance [109]. This metabolic heterogeneity influences the interplay between epithelial tumor cells and the TME, which particularly affects T cells and myeloid cells. One critical component of this interaction is the suppressive activity of Tregs, which are heavily reliant on the non-oxidative PPP for maintaining their function. Transketolase, a key enzyme in the non-oxidative PPP, is indispensable for Treg metabolism. Transketolase deficiency disrupts glycolysis, exacerbates oxidative stress, and leads to mitochondrial dysfunction due to excessive fatty acid and amino acid catabolism. Additionally, lower α-KG levels induce DNA hypermethylation, limiting the expression of genes essential for Treg suppressive activity. These findings underscore the pivotal role of the non-oxidative PPP in regulating Treg metabolism and immune function [110].

To further explore how epithelial tumor cells interact with the TME, we must consider how their distinct metabolic features impact both T cells and myeloid cells. Cholesterol plays a pivotal role in shaping immune responses within the TME by driving both T cell exhaustion and myeloid-derived suppressor cell (MDSC) activation. In CD8^+^ T cells, cholesterol accumulation triggers endoplasmic reticulum stress, which leads to activation of the UPR component XBP1. This pathway enhances the expression of immune checkpoints, including PD-1, 2B4, TIM-3, and LAG-3, diminishing T cell functionality and promoting exhaustion. CD8^+^ T cell antitumor activity is restored by targeting XBP1 or lowering cholesterol levels, offering a promising approach to reinvigorate T cell-based immunotherapy [111]. Simultaneously, in cancer cells, chronic activation of the UPR via XBP1 supports immune evasion through cholesterol synthesis and secretion. Cholesterol is transported via extracellular vesicles and internalized by MDSCs through macropinocytosis, enhancing their immunosuppressive activity. This cholesterol-driven MDSC activation further inhibits antitumor immunity.

Notably, therapeutic strategies targeting the XBP1/cholesterol axis or reducing cholesterol levels in tumors decrease MDSC abundance and reinstate robust antitumor responses, underscoring the potential of this pathway as a target to improve cancer immunotherapy outcomes [112]. In addition to cholesterol, ammonia is another key metabolite within the TME that profoundly impacts immune cell function and contributes to tumor immune evasion. For example, elevated ammonia levels in the CRC TME contribute to T cell dysfunction and resistance to immunotherapy. Using a metastatic mouse model, Bell et al. identified a robust accumulation of ammonia within tumors that reprograms T cell metabolism, promotes exhaustion, and reduces proliferation. Elevated serum ammonia levels and an ammonia-related gene signature in CRC patients correlate with impaired T cell responses, poor clinical outcomes, and ICI resistance. Moreover, enhanced ammonia detoxification reactivates T cells, reduces tumor growth, and improves anti-PD-L1 therapy efficacy [113]. The targeting of ammonia metabolism, such as enhancement of its clearance, could restore mitochondrial function and energy homeostasis in OXPHOS-driven tumors or reduce the metabolic burden in gluconeogenesis-reliant tumors. This dual approach may also improve the metabolic environment for T cells, enhancing immunotherapy efficacy.

Polyamine metabolism is another crucial player that is intricately connected to the one-carbon and folate pathways that support the metabolic demands of epithelial MSS CRC tumors reliant on OXPHOS and biosynthetic activities. Polyamine-blocking therapy (PBT), using α-difluoromethylornithine (DFMO) and a Trimer polyamine transport inhibitor, shows potential by targeting MDSCs and M2-like TAMs, key players in immunosuppression. Preclinical studies have revealed that PBT not only reduces tumor growth and metastasis but also enhances PD-1 blockade efficacy by increasing the number of tumor-specific cytotoxic T cells and prolonging survival [114]. Additionally, research in glioblastoma underscores the role of the arginine–ornithine–polyamine axis in sustaining tumor-associated myeloid cell (TAMC) survival in acidic microenvironments, a mechanism potentially relevant in CRC. DFMO disrupts this axis by lowering polyamine levels, impairing TAMC-mediated immunosuppression, and enhancing responses to immunotherapy and RT, making it a promising therapeutic approach for epithelial MSS CRC tumors [115].

## 6. Conclusions and Perspectives

The interplay between tumor metabolism and immune responses within the TME has emerged as a critical determinant of immune evasion and therapeutic resistance in CRC. Advances in our understanding of the metabolic adaptations of MSI and MSS subtypes have revealed distinct vulnerabilities that can be leveraged to enhance the efficacy of ICIs. These insights provide a foundation for innovative therapeutic strategies aimed at reshaping the TME and improving clinical outcomes in CRC.

MSI tumors are more responsive to ICIs due to the ability of these drugs to rewire metabolic dependencies, particularly glucose and glutamine, between cancer and immune cells. By targeting PD-1, ICIs can redirect these nutrients toward CD8^+^ T cells, enhancing their glycolysis and glutaminolysis while limiting tumor growth. This nutrient competition in MSI tumors underpins their immunogenic nature and highlights the potential of metabolic reprogramming to boost immune responses.

In contrast, the mesenchymal MSS CRC subtype represents a highly glycolytic and metabolically adaptable tumor type, leveraging pathways such as glycolysis and lactate metabolism to sustain growth and evade immune responses. This subtype produces immunosuppressive metabolites such as lactate and hyaluronan, which fuel Tregs, M2 macrophages, and exhausted CD8^+^ T cells, thereby contributing to immune resistance and therapy evasion. The interplay among tumor cells, T cells, macrophages, and the TME underscores the complexity of these immune-suppressive mechanisms. Emerging therapeutic strategies targeting key metabolic components, such as MCT4, hyaluronan, and ZEB1-mediated pathways, offer promising opportunities to disrupt these barriers and enhance the efficacy of ICIs in this aggressive tumor subtype.

On the other hand, the epithelial MSS CRC subtype exhibits diverse metabolic strategies, including reliance on gluconeogenesis, OXPHOS, and glycolysis, to adapt to the TME and evade immune responses. Key metabolites such as cholesterol, ammonia, and polyamines drive T cell exhaustion and enhance the immunosuppressive functions of MDSCs, TAMCs, and Tregs. These metabolic adaptations highlight therapeutic opportunities, including targeting the PPP, ammonia detoxification, and polyamine metabolism, to overcome immune resistance and enhance ICI efficacy.

Overall, these findings underscore the central role of metabolic adaptations in shaping the tumor–immune interplay in CRC. Approaches that address these metabolic vulnerabilities while leveraging the intrinsic immune potential of tumors such as MSI tumors hold significant promise for the development of more effective and tailored immunotherapeutic strategies.

Key metabolic singularities in MSS CRC include the mesenchymal subtype’s reliance on glycolysis and lactate metabolism, which supports immunosuppressive mechanisms involving Tregs and TAMs, and the epithelial subtype’s diverse use of OXPHOS, gluconeogenesis, and immunosuppressive metabolites such as cholesterol and polyamines. These metabolic features offer therapeutic opportunities, including targeting MCT4 and lactate pathways in mesenchymal MSS CRC and cholesterol metabolism, ammonia detoxification, and polyamine pathways in epithelial MSS CRC. Tailored therapeutic approaches that exploit these metabolic vulnerabilities have the potential to overcome immune resistance and improve the efficacy of immunotherapy in MSS CRC.

## Figures and Tables

**Figure 1 cancers-17-00498-f001:**
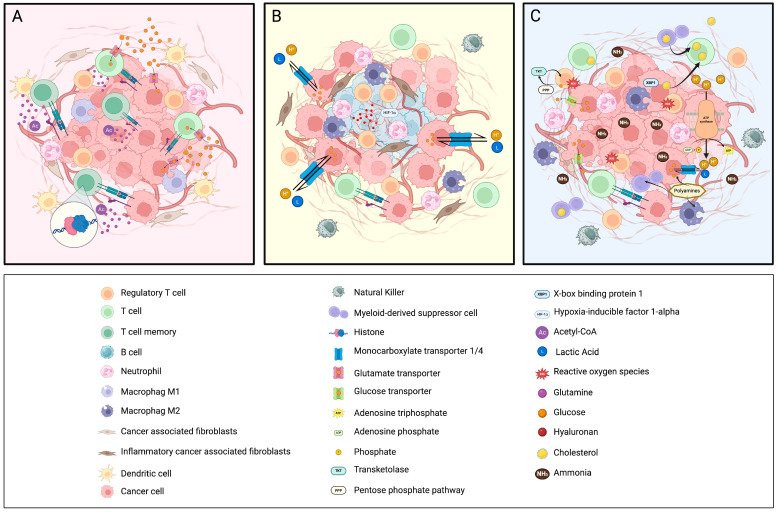
TME by immunometabolism subtype. (**A**) MSI: The tumor TME of MSI subtypes is characterized by active interactions between tumor cells and T cells. Competition for nutrients such as glucose and glutamine between tumor cells and T cells determines the prevailing metabolic pathways. These conditions render the TME more conducive, compared to other tumor subtypes, to immune cell infiltration, antigen presentation, and the establishment of functional immunological synapses. The generation of acetyl-CoA from glutamine metabolism supports the development of a memory phenotype in T cells and mitigates their exhaustion, thereby enhancing their functionality in antitumor immune responses. (**B**) MSS-Mesenchymal: The TME of mesenchymal tumors is characterized by a predominantly glycolytic metabolism with reduced OXPHOS, resulting in an excess of lactic acid. This acidic environment promotes dysfunction in effector T cells and NK cells while favoring a predominance of Tregs. Furthermore, the TME exhibits increased stromal stiffness due to the accumulation of hyaluronan, which exacerbates hypoxic conditions. This hypoxia activates iCAFs, neutrophils, and M2 macrophages. Collectively, these factors contribute to the establishment of a hostile and immunosuppressive environment. (**C**) MSS-Epithelial: The TME of the epithelial subtype exhibits a highly complex metabolism, where the tumor derives energy through glycolysis and OXPHOS. This metabolic activity creates a microenvironment enriched with metabolites such as cholesterol, produced by both the tumor and MDSCs, which leads to effector T cell dysfunction, a condition further exacerbated by the presence of ammonia. Lastly, the polyamine pathway is upregulated, fueling the tumor’s proliferative effects and enhancing the immunosuppressive activity of immune cells. (Created in BioRender; https://BioRender.com/s03i125, accessed on 27 January 2025).

**Table 1 cancers-17-00498-t001:** Prospective clinical trials with early-stage MSS CRC patients treated with ICIs.

Author	Number of Patients	Phase/ Primary Site	Schedule (Number of Cycles)	pCR, %	cCR or NOM, %
Rahma et al., 2021 [14]	185	II/R	FOLFOX (8) followed by LCRT+/−pembrolizumab (6) + S	31.9 vs. 29.4	-
Lin et al., 2024 [15]	231	III/R	SCRT or LCRT followed by CAPOX+/−camrelizumab (2) + S	39.8 vs. 15.3	-
Xiao et al., 2024 [16]	134	IIR/R	CAPOX+/−sintilimab (6) followed by LCRT + NOM or S	36.1 vs. 24.6	13.4 vs. 2.9
Gao et al., 2023 [17]	26	II/R	LCRT + tislelizumab (3) + S	50	-
Bando et al., 2022 [18]	37	II/R	LCRT + nivolumab (5) + S	30	-
Lin et al., 2021 [19]	30	II/R	SCRT followed by CAPOX + camrelizumab (2) + S	48	-
Grassi et al., 2023 [20]	60	II/R	LCRT followed by durvalumab (3) + S	32	-
Chalabi et al., 2020 [11]	20	II/C	Nivolumab (2) + ipilimumab (1) + S	11	-
Avallone et al., 2020 [12]	19	II/C	Nivolumab (2) + S	5	-
Pei et al., 2024 [13]	22	II/C	FOLFOX-B + sintilimab (6) + S	54	-

cCR, complete clinical response; S, surgery. R; rectal cancer. C; colon cancer.

**Table 3 cancers-17-00498-t003:** Quality of biomarkers using adapted REMARK criteria.

Author	Sample Size (*N*)	Biomarker	Prospective Design	Control Arm	Validation Set
Antoniotti et al., 2019 [23]	218	Immune-related 27-gene expression signature (DetermaIO) *	No	Yes	No
Thibaudin et al., 2023 [42]	57	TMB, intratumoral infiltration of CD8 cells, decorin	No	No	No
Lenz et al., 2023 [24]	310	TMB, CMS, *KRAS*	No	Yes	No
Cousin et al., 2021 [43]	48	TMB, PD-L1, intratumoral infiltration of CD8 cells	No	No	No
Parikh et al., 2021 [44]	40	TMB, CAFs, NK, EMT	No	No	No
Herting et al., 2021 [45]	24	LAG3, TNF/TGF-α, BTLA, VISTA, CD8^+^ T cells, CCL5, Flt3 ligand	No	No	No

* DetermaIO panel: APOD, ASPN, CCL5, CD52, COL2A1, CXCL11, CXCL13, DUSP5, FOXC1, GZMB, HTRA1, IDO1, IL23A, *ITM2A*, *KMO*, *KRT16*, *KYNU*, *MIA*, *PSMB9*, *PTGDS*, *RARRES3*, *RTP4*, *S100A8*, *SFRP1*, *SPTLC2*, *TNFAIP8*, and *TNFSF10.*

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
