# Peer review of "Metabolic Singularities in Microsatellite-Stable Colorectal Cancer: Identifying Key Players in Immunosuppression to Improve the Immunotherapy Response"

_cancers, 2025, doi:10.3390/cancers17030498_

Round 1
Reviewer 1 Report
Comments and Suggestions for Authors
Dear Authors
Review describes well about that - the interplay between tumor metabolism and immune responses within the tumor microenvironment (TME) has emerged as a critical determinant of immune evasion and therapeutic resistance in CRC.
The following steps should more clear information for readers to enjoy it
1) Please add two more keywords - tumor microenvironment (TME), microsatellite stable (MSS) CRC.
2) Add up-to-date references in the introduction section.
3) Please keep a simple and defined Graphical Abstract.
Author Response
Reviewer 1
Dear Authors
Review describes well about that - the interplay between tumor metabolism and immune responses within the tumor microenvironment (TME) has emerged as a critical determinant of immune evasion and therapeutic resistance in CRC.
The following steps should more clear information for readers to enjoy it
1) Please add two more keywords - tumor microenvironment (TME), microsatellite stable (MSS) CRC.
2) Add up-to-date references in the introduction section.
3) Please keep a simple and defined Graphical Abstract.
Response:
Dear Reviewer,
We sincerely thank you for your valuable evaluation of our manuscript and for recognizing the relevance of its content. In response to the minor changes requested, we have addressed the following points:
- We have added the two additional keywords you suggested, "tumor microenvironment" and "microsatellite stable." These can be found on page 1, highlighted in yellow, in the "Keywords" section (lines 42-43).
- As per your suggestion, we have updated the references in the Introduction section. On page 2, in the "Introduction" section (highlighted in yellow), we have included more recent epidemiological studies as well as a newer study on immunotherapy in CRC. These updates are located in lines 47, 52, and 60.
- We have created a graphical abstract as suggested, which depicts the most relevant interactions between the immune system and metabolism in colorectal cancer, clearly distinguishing between microsatellite unstable (MSI) and stable (MSS) tumors. This graphical abstract is attached as a PDF in the appropriate section, along with the BioRender publication license.
Reviewer 2 Report
Comments and Suggestions for Authors
Review Cancers 01/24
The review on „Metabolic singularities in microsatellite stable colorectal cancer: identifying key players in immunosuppression to improve the immunotherapy response” is well written and comprehensive. It covers clinical data on the role of immune checkpoint inhibition (ICI) alone or in combination in microsatellite stable (MSS) and microsatellite instable (MSI) colorectal cancer. A further section addresses data on predictive biomarkers in this context.
We would suggest that in table 1 and 2 the reference numbers of the citations should be directly included like in table 3 for the biomarker data.
In the following sections, which are in part quite long, the role of metabolism in ICI resistance is addressed and differences between the tumor microenvironments of MSI and MSS colorectal cancers are described in detail. In figure 1 it is not clear, what the illustration in the middle beyond the three pictures of the TME (A, B, C) is intended for (an epithelium and tumor cells?). The figure legend should be written in a way that the information in the figure can be understood without having read the main text. HIF-1alpha for example is not explained.
Finally, aspects of immune metabolism are covered in detail.
The heading 5.2 is a little bit misleading as in principle, more general information is given here, accounting for tumors in general and not specifically to MSI colorectal cancer as stated. We would suggest using subheadings to structure the text a little bit more especially in the long sections and it should be clear, which information is related to tumors in general vs. specifically related to MSS or MSI CRC.
Finally, it would be good to have a clear statement at the end, which are the key metabolic singularities in MSS CRC and how this could be therapeutically addressed. On page 15-17 some terms are in bold in the text.
Overall, this review provides a comprehensive overview on the topic, the literature covered is very well selected and the data are well described, thus providing many new insights for the reader.
Author Response
Reviewer 2
The review on Metabolic singularities in microsatellite stable colorectal cancer: identifying key players in immunosuppression to improve the immunotherapy response” is well written and comprehensive. It covers clinical data on the role of immune checkpoint inhibition (ICI) alone or in combination in microsatellite stable (MSS) and microsatellite instable (MSI) colorectal cancer. A further section addresses data on predictive biomarkers in this context.
We would suggest that in table 1 and 2 the reference numbers of the citations should be directly included like in table 3 for the biomarker data.
In the following sections, which are in part quite long, the role of metabolism in ICI resistance is addressed and differences between the tumor microenvironments of MSI and MSS colorectal cancers are described in detail. In figure 1 it is not clear, what the illustration in the middle beyond the three pictures of the TME (A, B, C) is intended for (an epithelium and tumor cells?). The figure legend should be written in a way that the information in the figure can be understood without having read the main text. HIF-1alpha for example is not explained.
Finally, aspects of immune metabolism are covered in detail.
The heading 5.2 is a little bit misleading as in principle, more general information is given here, accounting for tumors in general and not specifically to MSI colorectal cancer as stated. We would suggest using subheadings to structure the text a little bit more especially in the long sections and it should be clear, which information is related to tumors in general vs. specifically related to MSS or MSI CRC.
Finally, it would be good to have a clear statement at the end, which are the key metabolic singularities in MSS CRC and how this could be therapeutically addressed. On page 15-17 some terms are in bold in the text.
Overall, this review provides a comprehensive overview on the topic, the literature covered is very well selected and the data are well described, thus providing many new insights for the reader.
Response 2
Dear reviewer,
We sincerely appreciate your feedback and all the suggestions and changes you have proposed to improve the quality of this work.
Below, we will address each of your comments point by point:
1)” We would suggest that in table 1 and 2 the reference numbers of the citations should be directly included like in table 3 for the biomarker data”. Response: We have added references next to the citations in both Table 1 (page 3) and Table 2 (page 4), following the same format as in Table 3, as per your suggestion.
2) “In figure 1 it is not clear, what the illustration in the middle beyond the three pictures of the TME (A, B, C) is intended for (an epithelium and tumor cells?). The figure legend should be written in a way that the information in the figure can be understood without having read the main text. HIF-1alpha for example is not explained”. Response: Thank you for this clarifying point. As you suggested, we have removed the epithelium with the tumor, as it did not provide relevant information and could lead to confusion. Additionally, we have expanded and detailed the figure legend, including HIF-1alpha, to ensure that all relevant information is clear upon viewing the figure, as per your revision comments. The figure is located on page 12, and we have attached the updated BioRender license for its publication in this journal.
3) “The heading 5.2 is a little bit misleading as in principle, more general information is given here, accounting for tumors in general and not specifically to MSI colorectal cancer as stated. We would suggest using subheadings to structure the text a little bit more especially in the long sections and it should be clear, which information is related to tumors in general vs. specifically related to MSS or MSI CRC”. Response: Absolutely correct. We have changed the title of section 5.2 to "Metabolic Competition in the TME" (page 11, line 470) to address general concepts. Following this, we divided the text into two subsections for easier comprehension: 5.2.1 Metabolic Mechanisms Shaping Immune Responses (page 13, line 512) and 5.2.2 Therapeutic Implications in MSI and MSS CRC (page 13, line 549). In this way, as you suggested in your comment, the text is further specialized and made more fluid and understandable.
4) “Finally, it would be good to have a clear statement at the end, which are the key metabolic singularities in MSS CRC and how this could be therapeutically addressed”. Response: Absolutely agreed. We have added a final statement at the end of the last section, "Perspectives and Conclusions," to provide a clear and well-rounded conclusion to the review. It can be found on pages 19 and 20, from lines 876 to 884, highlighted in yellow.
5) “On page 15-17 some terms are in bold in the text”. Response: We have made the modification, and there are no longer any words in bold. Thank you for this correction.
Reviewer 3 Report
Comments and Suggestions for Authors
This review summarized the metabolism and immunosuppression in MSS colon cancers. MSS colon cancers usually do not respond to immunotherapy due to its specific TME. The mechanisms involved remain elusive. The review summarized the recent progress of the studies in the field. The review is well-written, well-organized, and well-structured. I enjoyed reading the whole manuscript. The paper can be accepted in present form.
Author Response
Reviewer 3
This review summarized the metabolism and immunosuppression in MSS colon cancers. MSS colon cancers usually do not respond to immunotherapy due to its specific TME. The mechanisms involved remain elusive. The review summarized the recent progress of the studies in the field. The review is well-written, well-organized, and well-structured. I enjoyed reading the whole manuscript. The paper can be accepted in present form.
Response 3
Thank you very much for your kind evaluation of the work carried out by our team in this review. Since no changes have been suggested and you propose that it can be accepted as it is, we have no additional comments to add.